# Coordinated Control of Intelligent Fuzzy Traffic Signal Based on Edge Computing Distribution

**DOI:** 10.3390/s22165953

**Published:** 2022-08-09

**Authors:** Chaodong Yu, Jian Chen, Geming Xia

**Affiliations:** School of Computer Science and Engineering, National University of Defense Technology, Changsha 410003, China

**Keywords:** ITS, edge computing, swarm intelligence, traffic signal control, fuzzy logic, offline learning

## Abstract

With the development of Internet of Things infrastructures and intelligent traffic systems, the traffic congestion that results from the continuous complexity of urban road networks and traffic saturation has a new solution. In this research, we propose a traffic signal control scenario based on edge computing. We also propose a chemical reaction–cooperative particle swarm optimization (CRO-CPSO) algorithm so that flexible traffic control is sunk to the edge. To implement short-term real-time vehicle waiting time prediction as a collaborative judgment of CRO-CPSO, we suggest a traffic flow prediction system based on fuzzy logic. In addition, we introduce a co-factor (collaborative factor) set based on offline learning to take into account the experiential characteristics of intersections in urban road networks for the generation of strategies by the algorithm. Furthermore, the real case of Changsha County is simulated on the SUMO simulation platform. The issue of traffic flow saturation is improved by our method. Compared with other methods, our algorithm enhances the proportions of vehicles that reach their destinations on time by 13.03%, which maximizes the driving experience for drivers. Meanwhile, our algorithm reduces the driving times of vehicles by 25.34%, thus alleviating traffic jams.

## 1. Introduction

With the rapid increase in the numbers of vehicles and a large and complex road network, delays, traffic accidents and environmental pollution due to vehicle queuing caused by traffic congestion have created an urgent need for traffic control strategies. How to improve traffic efficiency while reducing the traffic accident rate by using efficacious control measures has attracted the attention of both academia and industry [1,2,3,4]. Traffic signal control can effectively alleviate the saturated traffic conditions and improve the utilization of the road network. In recent years, many studies have focused on traffic signal control.

The earliest traffic control methods relied on hand signals [2]. In order to alleviate the economic losses caused by traffic congestion in Toronto, Hewton [1] proposed online optimization of traffic control signals through computers, which promoted the application of computer technology in the field of traffic management. For reasons of safety, Inose H [3] coordinated the timing of traffic signals according to the traffic flow on a road. With the continuous expansion of traffic scale, automatic traffic control strategies [4] have gradually formed a system. In the era of cloud computing, artificial intelligence (AI) algorithms continue to develop, and new concepts such as intelligent transportation systems (ITSs) have been put forward [5]. Wang [6] and Wu [7] et al. designed adaptive traffic signal control strategies by using deep reinforcement learning (DRL) of multi-agent cooperation to deal with large and complex road networks. Shao [8] took the weight of special vehicles into extra consideration when setting the state and reward function, and had an appropriate preference for special vehicles in traffic signal control. With the emergence of Internet of Vehicles (IoV) [9], edge computing [10,11,12] and 6G [13], vehicle-to-infrastructure (V2I) and traffic management mechanisms will make giant leaps. With the heterogeneity of the vehicle network and the further complexity of road structure, space–air–ground and even underwater vehicles may appear in the future [14], and road structure may tend to be multi-dimensional and multi-forking. Therefore, the proactive exploration of traffic signal control strategies is required. At present, many researches have carried out short-term traffic prediction based on the IoV [15,16,17,18,19], thus providing relatively reliable real-time predictions of changes in traffic conditions. In their study, Yang et al. [16] found that the total queuing time at traffic lights accounted for a large part of total driving time, which can be more than 50%. Thus, an efficient traffic signal control strategy is necessary. Wu et al. [20] thought that an intelligent algorithm was too expensive to be adapted to highly dynamic traffic flow when designing a traffic signal control strategy. The integration of edge computing and ITS [21] can sink the perception, collection and processing of traffic data to the edge. Through the reasonable deployment of an edge server [22], all the data on the traffic network can be interactive with low delay, making intelligent traffic signal control at the edge possible. Chen [23] applied an edge-distributed architecture combined with dynamic route guidance and signal control to effectively improve the traffic efficiency of the road network. However, in general, the above methods did not consider the use of regional traffic lights to achieve local coordinated regulation so as to achieve intelligent and efficient traffic flow scheduling. Moreover, traffic signal control using the architecture of edge computing needs further research.

Due to the existence of certain constraints and dynamic events (such as emergency vehicles and traffic evacuation, etc.), as well as subjective factors relating to drivers and pedestrians that need to be considered, uncertainty characterizes road traffic situations. Therefore, a fuzzy system [24] has been applied to effectively match the uncertainty of vehicles, traffic signals and control systems. Chou [25] simulated a traffic environment which is close to a real road network structure, and he designed a fuzzy logic controller suitable for this environment, which can achieve efficient reasoning with a few control rules. Gokulan [26] took limited environmental information as input and configured different levels of uncertainty in the rule base to realize a fuzzy logic reasoning system responding to traffic changes. Zaied [27] adjusted the time interval of traffic signals based on fuzzy logic decisions based on traffic conditions. He tested the system on real data sets and thereby increased the cycle speed of the system. The implementation of a fuzzy logic system is low cost, and we can achieve better inference effects through well-designed fuzzy rules. In this paper, a fuzzy logic system is used to predict traffic flow, and it is introduced into the fitness function of the traffic signal control algorithm to achieve global coordination evaluation.

The swarm intelligence algorithm [28] has also been studied in the context of solving the multi-dimensional nonlinear programming problem of traffic signal control. Ceylan [29] realized traffic signal timing optimization of balanced networks through genetic algorithms, thus improving computational efficiency and performance. Jose [30] optimized the traffic signal regulation of a large heterogeneous urban road network through a particle swarm optimization algorithm which reflected the potential effect of swarm intelligence algorithms in addressing such problems. Nasser [31] introduced an index scheme to control the iteration time of local optimization, thus accelerating the convergence speed of traffic signal timing strategy. Aleksandar [32] optimized the traffic control system of a large area in regional cities through the cluster principle of a bee colony collecting honey, thus improving the control range of a swarm intelligence algorithm for an urban road network. However, the swarm intelligence algorithm is still unable to cope with large-scale urban road network structures and the continuous complexity of traffic conditions. Moreover, how to construct a cooperative optimization framework to adapt to large-scale transportation network optimization is a problem that also needs further study.

Our study has investigated all of the research gaps mentioned above. We completely utilize the arithmetic power and control of edge scene, and we achieve cooperative traffic signal control based on fuzzy logic and the swarm intelligence algorithm. With this approach, it is possible to successfully address the issue of coordinated traffic scheduling for vast, intricate road networks, increasing traffic efficiency while enhancing the driving experience. Concretely, our main contributions are as follows:

Initially, we propose a traffic signal control scenario based on edge computing, and then we propose a new swarm intelligence algorithm: chemical reaction–cooperative particle swarm optimization (CRO-CPSO). We change the generation method of the traditional swarm intelligence strategy and make full use of the idle computing power at the edge to generate a local strategy so that the flexible traffic control sinks to the edge. The CRO-CPSO algorithm can effectively adapt to large-scale urban road network structures and complex and dynamic traffic conditions.

Secondly, we propose a traffic flow prediction system based on fuzzy logic and maintain the global traffic flow state table. We implement short-term real-time vehicle waiting time prediction as a collaborative judgment of CRO-CPSO, which effectively responds to the uncertainty of road traffic and achieves global coordination evaluation of the strategy.

Then, we introduce a co-factor (collaborative factor) set based on offline learning to take the experiential characteristics of intersections in an urban road network into account when generating strategies. The co-factor set integrates the potential influence of historical traffic flow of adjacent sections into the strategy generation, which increases the convergence of the algorithm and effectively adapts to the complexity of traffic, thus enhancing coordination in the regulation of large road network structures.

Finally, we examine the real case of Changsha County using the traffic simulator SUMO and compare other optimization algorithms (the evolutionary algorithm GA and the swarm intelligence algorithm PSO) to prove the effectiveness and advantages of CRO-CPSO.

The remainder of this paper is organized as follows. In Section 2, we describe our optimization methodologies, including an overall model diagram. In Section 3, we introduce the experimental methods and results. The conclusions and prospects for future work are detailed in Section 4.

## 2. Method

In this section, the specific details of the edge swarm intelligence ATSC environment (the scene of traffic signal control), the global traffic flow prediction (involved in collaborative judgment), and the CRO-CPSO model (the core of our solver technique) are introduced.

### 2.1. Edge Swarm Intelligence ATSC (Adaptive Traffic Signal Control) Environment

This subsection describes the structure and the data flow of the edge swarm intelligence ATSC environment.

Figure 1 shows the edge swarm intelligence ATSC environment, which includes cloud servers, MEC servers, traffic signal controllers, RSUs (roadside units) and vehicle units. The cloud layer carries on the global overall coordination. Cloud servers aggregate and generate the global co-factor set and predict the global traffic flow so as to make a collaborative judgment on the strategy of CRO-CPSO. In the MEC layer, the local strategy generation of CRO-CPSO is carried out, and the local co-factor set is trained off-line. MEC servers are deployed in their distribution area to achieve instantaneous data transmission with the traffic signal controllers. Vehicle units communicate with each other via V2V and exchange information with RSUs via V2I. The information is fed to MEC servers which communicate with each other via I2I. After aggregation and preprocessing by MEC servers, local road pheromone information is generated and summarized for cloud servers. The cloud servers perform fuzzy logic processing on the road pheromone information to obtain short-term real-time vehicle waiting time prediction as a collaborative judgment on the strategy of CRO-CPSO.

As shown in Figure 1, our algorithm is mainly executed in the MEC layer and cloud layer. Vehicle units transmit the road network data to the MEC layer. The MEC layer preprocesses road network data to generate local road pheromone information and upload it to the cloud layer. The MEC layer trains the road network data to generate local co-factor set, which is uploaded to the cloud layer. The cloud layer takes the road pheromone information as the input of the fuzzy logic system and obtains the short-term real-time vehicle waiting time prediction. The cloud layer performs offline training on road pheromone information and aggregates local co-factor sets to generate the global co-factor set. The cloud layer transmits the global co-factor set to the MEC layer. The MEC layer takes road pheromone information and the global co-factor set as the input for the swarm intelligence algorithm and obtains the local strategy of CRO-CPSO as the output. The cloud layer makes a collaborative judgment on the local strategy of CRO-CPSO through the short-term real-time vehicle waiting time prediction. Finally, the MEC layer transmits the optimal strategy of CRO-CPSO to the user layer.

### 2.2. The Global Traffic Flow Prediction

This subsection describes the pheromones in the global traffic flow prediction, the input and output fuzzy membership of the fuzzy logic system, and finally the update strategy for the states of vehicle units.

We designed a method to aggregate and process edge-distributed road pheromone information which can be used for traffic flow prediction. Short-term real-time vehicle waiting time prediction is used as the collaborative judgement on local traffic signal control strategies and provides support for local coordinated adjustment. In the edge-distributed ATSC environment, the underlying data constitute the inherent data set for a road, such as road structure names, identifications, geographical location and its graph topology. The dynamic data set of the road includes real-time vehicle speed and positional information and predictive vehicle routes. The data are transmitted to MEC servers via vehicle units and RSUs and are then further filtered and processed. The MEC servers obtain real-time speed and positional information from vehicle units and obtains inherent road data sets from RSUs. Then, MEC servers predict short-term real-time vehicle waiting time at the intersection and send the prediction to vehicle units. MEC servers share and coordinate with their neighboring servers via I2I.

Aggregating and generating global traffic flow predictions is a way to directly reflect the real-time state information of vehicle units in highly dynamic road networks. The pheromones used in global traffic flow prediction mainly include VRA (the current road name of the vehicle unit), VST (the time of vehicle unit switching state), VRN (the number of remaining roads for the vehicle unit) and so on. VRA can be obtained by analyzing the real-time location and speed of a vehicle unit and road map topology via V2I. When a vehicle unit runs on a road section at normal speed, then VRA is the name of this road section. When the real-time speed of the vehicle unit is lower than the normal speed, then VRA is “*waiting_state*”. The state of the vehicle unit mainly includes two states: moving and waiting. VST is mainly divided into two types: the remaining time from moving state to waiting state and the remaining time from waiting state to moving state. VST is mainly composed of experienced travel time, intersection delay caused by traffic signal control and delay caused by traffic congestion. VST can be calculated from the real-time vehicle unit speed and the road map topology obtained via V2I and the real-time queue at the intersection obtained via I2I. VRN is the prediction of the remaining path of the vehicle unit which can be obtained via V2I.

The first step is to calculate the experienced travel time. The speed information of vehicle units on the road network is updated frequently; thus, we calculate the real-time spatial average speed and use the real-time spatial average speed to calculate the experienced travel time. The real-time spatial average speed is the average speed of k samples within a time interval t, which can be calculated by Equation (1). Then, we divide the length of the road section travelled in time interval t by the real-time spatial average speed to obtain the experienced travel time, as shown in Equation (2).
(1)$${\underset{v}{¯}}_{V_{\sigma }}^{t}=\frac{\sum_{m=1}^{k}v_{V_{\sigma }}^{m}}{k}$$
(2)$$T_{V_{\sigma }}^{t}=L_{V_{\sigma }}^{t}/{\underset{v}{¯}}_{V_{\sigma }}^{t}$$
where $v_{V_{\sigma }}^{m}$ is the instantaneous speed of the mth sample of vehicle unit $V_{\sigma }$ and $L_{V_{\sigma }}^{t}$ is the length of the road section travelled by vehicle unit $V_{\sigma }$ in time interval t.

Then, we calculate the delay caused by traffic congestion. The traffic flow is a continuous cycle, and the subjective factors of drivers are uncontrollable. Any slight change in the traffic cycle may result in a mismatch between the calculated and optimal cycle. Thus we use fuzzy logic [33] in each successive cycle to overcome this mismatch. We construct the fuzzy system for the waiting time of vehicle units by establishing relations between the inputs and outputs of the fuzzy system using if–then rules. The proposed fuzzy logic system consists of four inputs: *VQL* (the queue length of vehicle units), *IWJ* (the judgement of whether to wait at the intersection), *GPD* (the duration of the green phase) and *VP* (the priority of vehicle units).

The *VQL* is the remaining queue length of vehicle units in the traffic flow. It contains three membership functions named *Zero*, *Short* and *Long* that range from 0 to 25 vehicle units, as illustrated in Figure 2a.

The *IWJ* is the judgement of whether a vehicle unit is waiting at the intersection. It contains three membership functions named *Ahead Oftime*, *Ordinary* and *Delayed* that range from 0% to 100%, as illustrated in Figure 2b.

The *GPD* is the duration of the green phase of the traffic light. It contains three membership functions named *Short*, *Medium* and *Long* that range from 0 to 60 s, as illustrated in Figure 2c.

The *VP* is the priority of vehicle units on the road network. It contains three membership functions named *Optimal*, *Suboptimal* and *Ordinary* that range from 0% to 100%, as illustrated in Figure 2d.

The output of the fuzzy logic system *WT* (the waiting time of a vehicle unit) is used to identify the delay caused by traffic congestion. It contains five membership functions called *RU* (Road Unobstructed), *LC* (Light Congestion), *MC* (Moderate Congestion), *SC* (Severe Congestion), *TU* (Traffic Tie-Up) that range from 0 to 100s, as illustrated in Figure 3. The proposed fuzzy logic system comprises forty-five fuzzy rules. Some of these rules are shown in Table 1, below.

The state of the vehicle unit needs to be updated constantly to ensure real-time and accurate representation of the current traffic status. The update strategy for the state of vehicle units is shown in Algorithm 1. With increase in $T_{g}$, *VST* decreases synchronously. When *VST* is non-zero, the vehicle unit will maintain the current state unchanged. When *VST* is zero, we need to judge whether the intersection of the current road section is the destination of the vehicle unit. When the vehicle unit does not reach the final destination, it will continue to drive. Thus, it is necessary to analyze the change of the state of the vehicle unit. If the vehicle unit is in “*waiting_state*”, then it ends the “*waiting_state*” and continues to drive on the next road section. *VST* will be reset and the experienced travel time for the next road section and intersection delay caused by traffic signal control will be added, as shown in Equation (3). If the vehicle unit was in the driving state before, then it arrives at the intersection. We need to judge whether the vehicle unit needs to wait for the traffic light. If the vehicle unit does not need to wait for the traffic light, it will pass through the intersection directly and continue driving on the next road section. *VST* will be reset and calculated, as shown in Equation (3). When the vehicle unit needs to wait for the traffic light, its *VRA* will be “*waiting_state*”. *VST* will be reset and WT will be added, as shown in Equation (4). When the vehicle unit arrives at its destination, we will no longer predict its real-time vehicle state, thus saving computing resources.
(3)$$VST=VST+T_{V_{\sigma }}^{t}+t_{\mathrm{delay}}$$
(4)VST=VST+WT
where $t_{\mathrm{delay}}$ is the intersection delay caused by traffic signal control.
**Algorithm 1** The update strategy of the state of vehicle unit1:**if** $T_{g}==0$ && VST>0 && VRA≠″waiting_state″ **then**2:  Analysis of vehicle unit queuing:3: **if** WT==0 **then**4:    Update *VRA* to name of next road section;5:    Reset *VST*;6:   $VST=VST+T_{e,i}^{t}+t_{\mathrm{delay}}$;7:   VRN=VRN−1;8: **else if** WT≠0 **then**9:   VRA=″waiting_state″;10:    Reset *VST*;11:   VST=VST+WT;12: **end if**13:**else if** VRA==″″waiting_state″ && WT==0 **then**14:  Update *VRA* to the name of next road section; Reset *VST*;15: $VST=VST+T_{e,i}^{t}+t_{\mathrm{delay}}$;16: VRN=VRN−1;17:**else if**WT==0 && VRN==0
**then**18: VRN=VRN−1;19:  Remove the vehicle unit from the predicting cycle;20:**end if**

### 2.3. CRO-CPSO Model

This subsection describes the fitness function, the co-factor set, the solution encoding and finally the optimization procedure for our proposed CRO-CPSO algorithm.

Research [29,30,31,32] on traffic signal control optimization has shown that swarm intelligence algorithms can outperform traditional methods in many cases. However, swarm intelligence algorithms have slow convergence speeds when dealing with multi-constraint optimization problems and cannot be well adapted to the current problem of large-scale and complex road networks. Thus, this paper proposes a distributed adaptive cooperative chemical reaction–cooperative particle swarm optimization (CRO-CPSO) algorithm. CRO-CPSO generates and iterates the local strategy at the edge via a distributed structure. It avoids the exponential growth of the solution space when confronted with large-scale road networks. The distributed structure also promotes cooperative control among traffic signal lights in the surrounding area. We use energy exchange as an indicator scheme to achieve an adaptive combination of local search and global exploration in the solution space. We also use a co-factor set to realize cooperative and coordinated control actions among adjacent edges and offline learning with historical traffic flow data.

When using the swarm intelligence algorithm to solve traffic signal control optimization, the scale of solution space will increase exponentially with the continuous expansion of a road network and the increasing complexity of intersections. Using the traditional centralized method, even if there is a powerful centralized cloud computing server, there will be a high computing cost and overhead. Thus, it cannot adapt to large-scale road networks. With the development of edge computing, the computing resources deployed on the edge have a wide application prospect. In our proposed CRO-CPSO, the local strategy for each intersection is generated using the computing resources of edge servers deployed at the intersections. It realizes the parallel utilization of distributed computing resources and reduces the computing load of the cloud server. The co-factor set is maintained on the adjacent edge server to realize offline learning of historical traffic data and cooperative control of adjacent traffic signals.

The traffic signal control optimization problem in this paper is a typical objective optimization problem. The solution space of the signal control set is divided into two parts: $S^{\prime }=\left\{{s^{\prime }}_{1},{s^{\prime }}_{2},\ldots ,{s^{\prime }}_{N_{\mathrm{ES}}}\right\}$ and $S\prime \prime =\left\{s{\prime \prime }_{1},s{\prime \prime }_{2},\ldots ,s{\prime \prime }_{N_{\mathrm{ES}}}\right\}$, where $S^{\prime }$ is the green phase sequence of each traffic light at the intersection, S″ is the duration of the green phase of the traffic light at the intersection and $N_{\mathrm{ES}}$ is the number of traffic lights at the intersection.

Considering information on various events during the simulation, the fitness function of CRO-CPSO is shown as Equation (5). Equations (8)–(10) are the constraints on it. The main objective of Equation (5) is to enhance the driver’s driving experience and reduce driving time. We achieve the incentive effect by increasing the number of vehicles that arrive at the destination within the reward time $T_{\Pi }$ and giving the reward score $D_{\mathrm{score}}$. We set the decision factor for the reward score (χ) so that the vehicle that runs out of time will not receive $D_{\mathrm{score}}$. Equation (8) ensures that each vehicle unit only passes through any intersection at most once. Equation (9) ensures that vehicle units with higher priority pass first while waiting at the intersection. Equation (10) ensures that only one traffic light is green at each intersection at any given time.
(5)$$\max\;Z\left(T\right)=\underset{V_{\sigma }\in N_{V}}{\sum }\chi \cdot \left[D_{\mathrm{score}}+T_{\Pi }-T_{V_{\sigma }}^{D}\right]$$
(6)$$T_{V_{\sigma }}^{D}=\int^{\;}\sum_{\vartheta \in \Theta_{V_{\sigma }}}T_{V_{\sigma },\vartheta }^{t}\mathrm{dt}+\sum_{\delta \in \Xi_{V_{\sigma }}}{\mathrm{WT}}_{V_{\sigma },\delta }+\left(\Theta_{V_{\sigma }}-1\right)\cdot t_{\mathrm{delay}}$$
(7)$$\chi =\{\begin{array}{c} 0,\;{\;T}_{V_{\sigma }}^{D}\geq T_{\Pi }\\ 1,\;{\;T}_{V_{\sigma }}^{D}<T_{\Pi } \end{array}$$
where $D_{\mathrm{score}}$ is the reward score, $T_{\Pi }$ is the maximum time to ensure the drivers’ driving experience, $N_{V}$ is the set of all vehicle units, $\Theta_{V_{\sigma }}$ is the set of road sections of vehicle unit $V_{\sigma }$ and $\Xi_{V_{\sigma }}$ is the set of wating state of vehicle unit $V_{\sigma }$.
(8)$$\{\begin{array}{c} n_{V_{\sigma }}^{\left(i\right)}=1,\;\forall i,j\in \Phi_{V_{\sigma }}\;,\;\varphi_{i,j}\in \Psi_{V_{\sigma }}\\ n_{V_{\sigma }}^{\left(i\right)}=0,\;\forall i,j\in \Phi -\Phi_{V_{\sigma }}\;,\;\varphi_{i,j}\in \Psi -\Psi_{V_{\sigma }} \end{array}\;$$
where $n_{V_{\sigma }}^{\left(i\right)}$ is the number of times vehicle unit $V_{\sigma }$ arrives at intersection i, Φ is the set of all intersections, $\varphi_{i,j}$ is the road section starting from intersection i and ending at intersection j, and Ψ is the set of all road sections.
(9)$$\varpi_{V_{\sigma }}^{\varphi_{i,j}}<\varpi_{V_{\sigma }{}^{\prime }}^{\varphi_{i,j}}\;,{\;\mathrm{VP}}_{V_{\sigma }}<{\mathrm{VP}}_{V_{\sigma }{}^{\prime }},\forall {\mathrm{VP}}_{V_{\sigma }},{\mathrm{VP}}_{V_{\sigma }{}^{\prime }}\in {\mathrm{VP}}_{t,\varphi_{i,j}}^{\left(j\right)}$$
where $\varpi_{V_{\sigma }}^{\varphi_{i,j}}$ is the queuing sequence of vehicle unit $V_{\sigma }$ waiting for the green light in road section $\varphi_{i,j}$ when it arrives at intersection j and ${\mathrm{VP}}_{t,\varphi_{i,j}}^{\left(j\right)}$ is the set *VP* of vehicle units arriving at intersection j in road section $\varphi_{i,j}$ in time interval t.
(10)$$\{\begin{array}{c} \underset{j}{\sum }\Omega_{\varphi_{i,j}}=1,\;\forall t\cap^{\;}\forall j\in \Phi_{\to j}\\ \Omega_{\varphi_{i,j}}\in \left\{0,1\right\} \end{array}$$
where $\Omega_{\varphi_{i,j}}$ is the identifier of the state of the green traffic light in road section $\varphi_{i,j}$ and $\Phi_{\to j}$ is the set of all intersections leading to intersection j.

In this paper, we use a co-factor set to realize the cooperative and coordinated control actions among adjacent edges and offline learning with historical traffic flow data. The co-factor set objectively reflects the congestion of each road section and the potential “*Attacking Traffic Flow*” (after passing the intersection, the vehicle units will enter the next road section, thus increasing the congestion level and attacking the traffic density of the road section) among road sections. As the experienced set of global road sections, the co-factor set reflects both the past traffic flow information of each intersection and the associated influence among road sections. Thus, we can deepen the degree of cooperation among edge servers in the global control of traffic lights by introducing the co-factor set. It enables edge servers to consider the cooperation among servers in generating local strategies. The joint reward feedback for each edge server will further iterate the co-factor set, so as to take experience and timeliness into account.

As the first step, we calculate the joint reward feedback for each edge server. We take the average waiting time of vehicle units within the coverage area of edge servers as the evaluation index. ${\bar{\mathrm{WT}}}_{t}^{\kappa }$ is the average waiting time of vehicle units within the coverage area of edge server κ in time interval t, as shown in Equation (11). $r_{\kappa ,\mathrm{local}}$ is the local reward feedback for edge server κ, as shown in Equation (12). The adjacent edge servers have traffic flow correlation. Thus, a spatial attenuation factor is introduced so that the joint reward feedback can reflect the gains of the adjacent environment. $r_{\kappa ,\mathrm{joint}}$ is the joint reward feedback for edge server κ, as shown in Equation (13). $\rho_{t}^{\kappa }$ is the traffic density within the coverage area of edge server κ in time interval t, as shown in Equation (14).
(11)$${\bar{\mathrm{WT}}}_{t}^{\kappa }=\frac{\sum_{N_{V}^{\kappa }}{\mathrm{WT}}_{\kappa }}{N_{\mathrm{WT}}^{\kappa }},\forall \kappa \in N_{\mathrm{ES}}$$
where $N_{V}^{\kappa }$ is the set of all vehicle units within the coverage area of edge server κ, $N_{\mathrm{WT}}^{\kappa }$ is the set *WT* of the vehicle units within the coverage area of edge server κ and $N_{\mathrm{ES}}$ is the set of all the edge servers.
(12)$$r_{\kappa ,\mathrm{local}}=\{\begin{array}{c} 1\;\mathrm{if}\;{\bar{\mathrm{WT}}}_{t}^{\kappa }>{\bar{\mathrm{WT}}}_{t-1}^{\kappa }\;\\ 0\;\mathrm{if}\;{\bar{\mathrm{WT}}}_{t}^{\kappa }={\bar{\mathrm{WT}}}_{t-1}^{\kappa }\\ -1\;\mathrm{if}\;{\bar{\mathrm{WT}}}_{t}^{\kappa }<{\bar{\mathrm{WT}}}_{t-1}^{\kappa } \end{array}$$
(13)$$r_{\kappa ,\mathrm{joint}}=r_{\kappa ,\mathrm{local}}+\underset{{Κ}_{\kappa }}{\sum }\frac{1}{\left|d_{\kappa ↔{Κ}_{\kappa }}\right|}\times r_{{Κ}_{\kappa },\mathrm{local}}$$
where $\frac{1}{\left|d_{\kappa ↔{Κ}_{\kappa }}\right|}$ is the spatial attenuation factor and ${Κ}_{\kappa }$ is the set of the adjacent edge servers of edge server κ. Due to the fast attenuation of space, the set of adjacent edge servers only consider a two-layer road network structure.
(14)$$\rho_{t}^{\kappa }=\frac{\sum_{\varphi \in \Psi_{\kappa }}N_{\varphi ,t}^{\kappa }}{\sum_{\varphi \in \Psi_{\kappa }}L_{\varphi }^{\kappa }}$$
where $\Psi_{\kappa }$ is the set of all the road sections within the coverage area of edge server κ, $N_{\varphi ,t}^{\kappa }$ is the number of vehicle units on the road section φ within the coverage area of edge server κ in time interval t and $L_{\varphi }^{\kappa }$ is the length of the road section φ within the coverage area of edge server κ.

Then we construct the co-factor set A as a matrix of Κ×Κ, as shown in Equation (15). The co-factor set is acquired through offline learning and iterative updating with immediate joint reward feedback, as shown in Equation (16).
(15)$$A=\left[\begin{array}{ccc} A_{11} & \cdots & A_{1Κ}\\ \vdots & \ddots & \vdots \\ A_{Κ1} & \cdots & A_{ΚΚ} \end{array}\right]$$
(16)$$A_{\mathrm{ij}}=A_{\mathrm{ij}}+\alpha \left(r_{\mathrm{ij},\mathrm{joint}}+{\gamma \rho }_{t}^{\mathrm{ij}}+\frac{d_{\epsilon }}{\left|d_{\mathrm{ij}}\right|}\frac{\rho_{t}^{i}-\rho_{t}^{j}}{\rho_{t}^{j}}\right),\forall i,j\in Κ$$
where $d_{\epsilon }$ is the benchmark distance, $\left|d_{\mathrm{ij}}\right|$ is the distance between edge server i and edge server j, and α and γ are the attenuation factors.

The swarm intelligence algorithm can be used to solve the multi-objective optimization problem of traffic signal control. In the traditional swarm intelligence algorithm, each iteration of the solution space applies a local search algorithm, which requires a lot of computing resources and reduces the convergence efficiency. Edge servers only have limited computing resources and storage capacities. Thus, we adopt the optimization framework of the chemical reaction optimization algorithm [34] to accelerate the convergence of traffic signal control optimization strategies. We adopt the PSO [35] algorithm with excellent inter-individual coordination and introduce the co-factor set under the CRO framework to realize local cooperative scheduling and global control among traffic lights in the multi-dimensional control problem of large-scale traffic light control. By configuring two necessary attributes, molecular potential energy (PE) and molecular kinetic energy (KE), the algorithm can avoid falling into local optima too early and converge to optima faster. PE represents the stability of the solution space and is defined as the reciprocal of vehicle travel time, as shown in Equation (17). When PE increases, the solution space tends to be stable. Thus, the global exploration will be stopped and local searching will be performed. KE makes the solution space tend to be dynamic. When KE is high, the global exploration will be continued to avoid falling into local optima too early. Therefore, only when KE attenuates to a threshold value and PE tends to be stable, will local searching be carried out, thus greatly improving the convergence efficiency. Thus, the algorithm can be deployed on edge servers with limited computing resources.
(17)$$\mathrm{PE}=\frac{1}{\sum_{V_{\sigma }\in N_{V}}T_{V_{\sigma }}^{D}}$$

The pseudocode of CRO-CPSO is shown in Algorithm 2. The input of Algorithm 2 is road pheromone information and the global co-factor set; its output is the local traffic light regulation strategy. The solution space $S{}^{\prime }=\left\{{s{}^{\prime }}_{1},{s{}^{\prime }}_{2},\ldots ,{s{}^{\prime }}_{N_{\mathrm{ES}}}\right\}$ is the green phase sequence of each traffic light at the intersection, where $\left\{{s{}^{\prime }}_{1},{s{}^{\prime }}_{2},\ldots ,{s{}^{\prime }}_{N_{\mathrm{ES}}}\right\}$ is an array of different integers ranging from 1 to $N_{\mathrm{ES}}$. In this paper, we use Monomolecular Decomposition and Polymolecular Synthesis to run the global exploration of the solution space. When the solution space tends to be stable, we use Monomolecular Collision and Polymolecular Collision to run the local search in the solution space. Figure 4 shows the updating scheme for the solution space $S{}^{\prime }$ of dimension 6.
**Algorithm 2** Pseudocode of CRO-CPSO1:InitialSwarm()2:m←13:**while** m<MaxIter **do**4:  **for** each particle $x_{ij}^{m}$ **do**5:   **if** $Self-Collision\left(S{}^{\prime }\right)$ **then**6:
    $status\leftarrow Monomolecular\;Collision\left(S{}^{\prime }\right)$
7:   **else if** $Self-Decomposition\left(S{}^{\prime }\right)$ **then**8:
    $status\leftarrow Monomolecular\;Decomposition\left(S{}^{\prime }\right)$
9:   **else if** $Intergroup-Collision\left(S{}^{\prime }\right)$ **then**10:
    $status\leftarrow Polymolecular\;Collision\left(S{}^{\prime }\right)$
11:   **else if** $Intergroup-Synthesis\left(S{}^{\prime }\right)$ **then**12:
    $status\leftarrow Polymolecular\;Synthesis\left(S{}^{\prime }\right)$
13:
   **end if**
14:
   $v_{ij+1}^{m}=updateVelocity\left(\omega ,v_{ij}^{m},c_{1},c_{2},E_{ij}^{m},Q_{ij}^{m},x_{ij}^{m}\right)$
15:
    $x_{ij+1}^{m}=updatePosition\left(v_{ij+1}^{m},x_{ij}^{m},A_{k}\right)$
16:
   $\mathrm{fitness}\left(S{}^{\prime },S\prime \prime \right)$
17:    Update $E_{ij}^{m}$18:    Update $Q_{ij}^{m}$19:
  **end for**
20:**end while**

When the solution space satisfies the condition of *S**elf-C**ollision*, Monomolecular Collision will occur. As shown in Figure 4a, the solution space is slightly changed. We use the method of Swapping Two-Domain Spaces to ensure that the spatial arrangement of S’ is not repetitive. After the Collision, the KE of S’ decays and the PE of S’ is updated. When the solution space satisfies the condition of *Self-D**ecomposition*, Monomolecular Decomposition will occur. As shown in Figure 4b, the solution space is changed considerably. Select a breakpoint for S and divide it into two parts [first_half , second_half]. S1 retains [first_half] and S2 retains [second_half]. Then, the remaining parts of S1 and S2 are generated while ensuring that the arrangement of solution spaces is not repetitive. The KE between the solution spaces is redistributed and the PE of the solution spaces is updated.

When the solution space satisfies the condition of *I**ntergroup-C**ollision*, Polymolecular Collision will occur. As shown in Figure 4c, the solution space is slightly changed. In the Polymolecular Collision, we use *C**onflict-D**etection* to map the conflicting values so as to ensure the non-repeatability of the arrangement between S1′ and S2′. In the example of *C**onflict-D**etection* in Figure 4c, we can see two sets of mappings 2→5→4 and 3→1. Thus, if there are two 2 s in the solution space after the Polymolecular Collision, 2 will be converted to 4, and so on, until there is no conflict. After the Collision, the KE between the solution spaces is redistributed and the PE of the solution spaces is updated. When the solution space satisfies the condition of *Intergroup-S**ynthesis*, Polymolecular Synthesis will occur. As shown in Figure 4d, the solution space is changed considerably. Choose a synthesis point for the two solution spaces. Then, the [first_half] of S1 and the [second_half] of S2 are combined to produce a new solution space S with great diversity. In the Polymolecular Synthesis, we also use *C**onflict-D**etection* to map the conflicting values so as to ensure the non-repeatability of the arrangement of S. After the synthesis, the KE of the solution spaces is aggregated, and the PE of the solution space is updated.

The solution space $S\prime \prime =\left\{s{\prime \prime }_{1},s{\prime \prime }_{2},\ldots ,s{\prime \prime }_{N_{\mathrm{ES}}}\right\}$ is the duration of the green phase of a traffic light at an intersection, where $\left\{s{\prime \prime }_{1},s{\prime \prime }_{2},\ldots ,s{\prime \prime }_{N_{\mathrm{ES}}}\right\}$ is an array of different integers ranging from 0 to 60 s. When edge server i generates the duration of the green phase of the traffic light, we introduce co-factor set A into the solution space. We focus on the k-neighbor road sections that are directly related to the current intersection $\Phi_{i}$ in updating the solution space. The co-factor $A_{\mathrm{ij}}$ reflects the potential “*Attacking Traffic Flow*” (including the potential traffic flow of other road sections extending from intersection $\Phi_{j}$) at intersection $\Phi_{i}$ in the direction of road section $\varphi_{j\to i}$. Thus, when $A_{\mathrm{ij}}$ accounts for a large proportion of $A_{k}$, it indicates that the road section $\varphi_{j\to i}$ may become congested. Thus, experience-based coordinated regulation can be implemented to alleviate the potential congestion of road sections. We need to extend the green phase of road section $\varphi_{j\to i}$ in an appropriate proportion within the green cycle. The green phases of other road sections with smaller proportions in $A_{k}$ are appropriately compressed. The ratio of extension to compression is determined by the weight ratio. The extension–compression ratio is also affected by the experiential factor due to the empirical lag of the co-factor set, as shown in Equation (18).

We introduce the idea of particle swarm optimization in the iteration of $S\prime \prime =\left\{s{\prime \prime }_{1},s{\prime \prime }_{2},\ldots ,s{\prime \prime }_{N_{\mathrm{ES}}}\right\}$. Each potential solution to the problem is the position of the particle, and the particles are updated iteratively on a population scale. The particles are initialized before the iteration begins. The fitness of the particles is calculated using Equation (5) as the initial $E_{\mathrm{ij}}^{m}$ and $Q_{\mathrm{ij}}^{m}$ of the particles. Then, the circular heuristic search process is initiated: $v_{\mathrm{ij}+1}^{m}$ and $x_{\mathrm{ij}+1}^{m}$ ($x_{\mathrm{ij}+1}^{m}$ for the particle is rounded in the updating process) for the particle are updated iteratively, and its fitness is calculated. If the fitness is better than $E_{\mathrm{ij}}^{m}$, update $E_{\mathrm{ij}}^{m}$. If the fitness is better than $Q_{\mathrm{ij}}^{m}$, update $Q_{\mathrm{ij}}^{m}$. In each iteration update, the particle updates its position according to Equation (18):(18)$$x_{\mathrm{ij}+1}^{m}=\{\begin{array}{c} \lceil x_{\mathrm{ij}}^{m}+v_{\mathrm{ij}+1}^{m}+\beta \cdot \left(\frac{\sum_{X}x_{\mathrm{ij}}^{m}\cdot A_{\mathrm{ij}}}{\sum_{A_{k}}A_{\mathrm{ij}}}-x_{\mathrm{ij}}^{m}\right)\rceil \;\mathrm{if}\;\frac{\sum_{X}x_{\mathrm{ij}}^{m}\cdot A_{\mathrm{ij}}}{\sum_{A_{k}}A_{\mathrm{ij}}}\geq x_{\mathrm{ij}}^{m}\\ \lfloor x_{\mathrm{ij}}^{m}+v_{\mathrm{ij}+1}^{m}+\beta \cdot \left(\frac{\sum_{X}x_{\mathrm{ij}}^{m}\cdot A_{\mathrm{ij}}}{\sum_{A_{k}}A_{\mathrm{ij}}}-x_{\mathrm{ij}}^{m}\right)\rfloor \;\mathrm{otherwise}. \end{array}$$
where β is the experiential factor, ⌈ ⌉ and ⌊ ⌋ represent rounding processing and $v_{\mathrm{ij}+1}^{m}$ is the velocity of the particle, as shown in Equation (19):(19)$$v_{\mathrm{ij}+1}^{m}=\omega \cdot v_{\mathrm{ij}}^{m}+c_{1}\cdot U\left(0,1\right)\cdot \left(E_{\mathrm{ij}}^{m}-x_{\mathrm{ij}}^{m}\right)+c_{2}\cdot U\left(0,1\right)\cdot \left(Q_{\mathrm{ij}}^{m}-x_{\mathrm{ij}}^{m}\right)$$
where $E_{\mathrm{ij}}^{m}$ is the individual extremum, $Q_{\mathrm{ij}}^{m}$ is the global extremum, $c_{1}$ and $c_{2}$ are learning factors, U(0,1) is a uniform random value in [0, 1] and ω is the inertia weight, as shown in Equation (20):(20)$$\omega =\omega_{\max}-\frac{\left(\omega_{\max}-\omega_{\min}\right)\cdot P}{P_{\max}}$$
where $\omega_{\max}$ is the maximum of the inertia weight, $\omega_{\min}$ is the minimum of the inertia weight and $P_{\max}$ is maximum iteration.

## 3. Experimental Setup

This section presents the experimental framework followed to assess the performance of our method. We first describe the specific road network scenario generated for this paper. Then, we present the detailed simulation parameters.

We used MATLAB to achieve the CRO-CPSO algorithm. The simulation phase was carried out by executing the traffic simulator SUMO v1.12.0 ("Microscopic Traffic Simulation using SUMO"; Pablo Alvarez Lopez, Michael Behrisch, Laura Bieker-Walz, Jakob Erdmann, Yun-Pang Flötteröd, Robert Hilbrich, Leonhard Lücken, Johannes Rummel, Peter Wagner, and Evamarie Wießner. IEEE Intelligent Transportation Systems Conference (ITSC), 2018.) for Windows.

SUMO [36] is a well-known traffic simulator. It provides an open source, microscopic, multi-modal traffic simulation environment so as to realize the simulation of traffic signal scheduling of large road network structures. We imported digital maps from OpenStreetMap (OSM) [37]. The digital map was then transformed and combed through to provide a valid SUMO network using the netconvert script provided in the SUMO package.

We generated an actual road network scenario from the real digital map, as shown in Figure 5. The physical location is Changsha County, Changsha City, Hunan Province, China, which has 64 road intersections. We choose this area because Changsha County has a regular road network structure which can represent the general road network structure found in the central urban area of China and because examining a real case has more practical significance in the context of solving road congestion problems. As for traffic density, this paper tested three different density levels (low density: 300 vehicles, medium density: 500 vehicles, high density: 1000 vehicles) to consider the road traffic flow situations in different periods of time.

In the CRO-CPSO algorithm, we set the swarm (population) size to 30 particles and the number of iterations to 100 steps. We set other parameters of the algorithm based on a small area of Changsha County (with 64 traffic lights and 100 vehicles). The detailed simulation parameters are shown in Table 2.

Additionally, we implemented two algorithms, GA [29] and PSO [35], in order to establish comparisons against our CRO-CPSO algorithm. The GA algorithm is a classical algorithm used in evolutionary computation, which mainly imitates the survival of the fittest in nature to carry out natural evolutions. The PSO algorithm is a classical algorithm used in swarm intelligence, which mainly searches for optimizations through a heuristic search process. GA and PSO have good applications in solving nonlinear programming problems, so we selected them as comparison algorithms.

## 4. Experiment and Analysis

This section presents the results and analyses from several viewpoints. First, we studied and analyzed the influence of parameters on the performance of the algorithm. Then we conducted comparative experiments. Finally, we analyzed the scalability of the algorithm.

### 4.1. Performance Analysis of Algorithms

Before carrying out the comparative experiment, we first analyzed the performance of the CRO-CPSO algorithm with different swarm sizes and maximum numbers of iterations through a series of experiments. We set the parameter values in the following experiment by this investigation. Considering a small-scale case of Changsha County (with 64 traffic lights and 100 vehicles), we tried different configuration combinations and plotted the traces of the progress of the best fitness values, as shown in Figure 6. These traces corresponded to the configuration combinations of swarm sizes with 10, 20 and 30 particles and maximum iterations of 50, 100 and 200 steps. The swarm size will affect the diversity of the population and the maximum number of iterations will affect the inertia weight; thus, we mainly studied the parameter values through their different configuration combinations.

As shown in Figure 6, for all configuration combinations of swarm sizes and maximum numbers of iterations, our CRO-CPSO algorithm can converge within the interval of 50 to 100 iterations. The algorithm achieved the best performance results under the configuration combination of a swarm size of 30 particles and a maximum iteration of 100 steps. We found that when the maximum iterations was 50 steps, due to the small number of iterations, even if the swarm size was large, it was easier to converge to the local optimal value (between 7.2 × 10^4^ and 7.25 × 10^4^). When the swarm size was 10 particles, the small swarm size limited the evolutionary diversity of the population. Thus, even when the maximum iterations was 200 steps, the population could only converge to a low local optimal value (7.37 × 10^4^). Therefore, when choosing a configuration combination, one needs to consider both computational cost and optimal fitness value. We found that when the swarm size was 30 particles and the maximum iterations was 100 steps, the population could converge to an approximately optimal value. Although the population achieved a higher fitness value under the configuration combination of a swarm size with 30 particles and maximum iterations of 200 steps, the small increase in fitness value (0.015 × 10^4^) was not matched by the expensive computation cost (3000 function evaluations). Therefore, we opted to set the swarm size as 30 particles and the maximum iterations as 100 steps in our experimentation.

In Figure 7, we plotted the fitness distribution of the whole population in the optimization process of the CRO-CPSO algorithm. Specifically, the plot mainly illustrates the operation of the algorithm in the case of Changsha County (with 64 traffic lights and 300 vehicles). We can see that in the early stages the particles were diverse and with low and fluctuating fitness regions (between 1 and 3). Then they gradually converged to a higher fitness with constant iterations. Thus, all the particles showed ideal convergence and robustness in spite of the differences among them. In this optimization process, 277 vehicles out of 300 arrived at the destination within maximum time $T_{\Pi }$ (92.3% of the vehicles have high-quality driving experience), as shown in Figure 8. We found that under the baseline control strategy, 37 vehicles did not have high-quality driving experience (i.e., they were unable to reach the destination within $T_{\Pi }$), and the journey time of all vehicles was generally high. CRO-CPSO introduces the co-factor set; thus, the influence of potential traffic flow on road conditions is considered. The collaborative effect will not only ensure the driving experience of vehicles but also appropriately extend the waiting time in unobstructed road sections and shorten the waiting time in crowded road sections so as to achieve the effect of alleviating traffic congestion. We introduced the reward score $D_{\mathrm{score}}$ and $T_{\Pi }$ as the incentive mechanism in the calculation of the fitness function so that the convergence of the algorithm takes into account the driving experience of vehicles. It can be seen from Figure 8 that under the CRO-CPSO control strategy, the overall driving time of 300 vehicles is reduced by 11,937 s (about 199 min) compared with the baseline control strategy, thus reflecting the effectiveness of this algorithm in improving the traffic efficiency of the urban road network. We found that due to traffic light control, the driving times of a few vehicles with low driving times increased slightly (i.e., waiting time in unobstructed road sections was extended), and the driving times of some vehicles with high driving times was sharply reduced (i.e., waiting time in crowded road sections was shortened), which verifies the coordinated control of the co-factor set for each intersection of the urban road network. We found that the number of vehicles with low-quality driving experience decreased by 14 (i.e., a 37.8% reduction), and no single vehicle had too excessive a driving time (the longest vehicle driving time under the baseline strategy was also reduced from 1698 s to 1384 s after optimization). As each iteration of the algorithm will take the driving experience of the vehicles as the incentive factor, it can well avoid the extreme phenomenon of shortening the waiting time of the majority of vehicles by seriously affecting the driving experience of a very few vehicles. Thus, the CRO-CPSO algorithm takes into account the driving experience of almost all vehicles while ensuring the optimum efficiency.

### 4.2. Comparative Experimental Analysis

In this section, we present a test and comparison of three algorithms, our proposed CRO-CPSO, GA and PSO, in the case of Changsha County. Compared with GA, we expected to prove the advantages of using particle swarm optimization (PSO) over evolutionary algorithms in urban traffic control problems. By comparison with the traditional PSO algorithm, we expected to prove that the introduction of prior knowledge (short-term traffic prediction and co-factor set) in the swarm intelligence algorithm could improve the performance of road cooperative regulation. In this case, we set three traffic flow densities (100 vehicles, 300 vehicles and 500 vehicles) to simulate different congestion conditions on the urban road network.

Table 3 contains the best fitness values for CRO-CPSO, GA and PSO for different traffic flow densities in Changsha County. We found that with the increase in traffic flow density, the best fitness values of the three algorithms all showed a nonlinear increase in different proportions. Through an analysis of the traffic flow data set, we found that although the number of vehicles increased proportionally, the track of each vehicle was randomly generated, which affected the waiting time of each vehicle. The number of vehicles increased greatly; thus, the best fitness values for the algorithm showed an overall upward trend. We found that the best fitness values for CRO-CPSO were better than those for GA and PSO under different traffic flow densities. We also found an interesting feature. The best fitness values for GA were better than those for PSO at low traffic flow densities (100 vehicles, 300 vehicles). However, when the urban road network entered a high-congestion state (500 vehicles), the best fitness value for PSO exceeded that for GA. Therefore, the PSO algorithm is more suitable than the GA algorithm when dealing with the congestion typical of a complex urban road network. We also found that with the increase in traffic flow density, the performance improvements of CRO-CPSO with regard to traffic control were 2.70% (100 vehicles), 4.10% (300 vehicles) and 13.03% (500 vehicles), respectively. CRO-CPSO introduces co-factor sets and makes collaborative decisions based on fuzzy logic. When congestion occurs in urban road networks, the coordinated regulation can evacuate potentially congested road sections in a targeted way (that is, the green time of traffic lights can be appropriately extended/compressed in a coordinated way). Therefore, when regulating urban road networks with high congestion, our proposed CRO-CPSO shows a better performance than the other algorithms.

In Figure 9, we can see the distribution of the journey times of vehicles under the control of the three algorithms with different traffic flow densities on the urban road network. We found that under the three traffic flow densities, the median journey time of vehicles regulated by CRO-CPSO was the lowest, which reflects the excellent performance of our proposed CRO-CPSO in improving the traffic efficiency of the urban road network. The box height of CRO-CPSO was the lowest, which reflects the low fluctuation in the journeys of vehicles. As the traffic flow density reached 500 vehicles, congestion began to occur on the road network (both the maximums for the journey times of vehicles and the numbers of outliers in GA and PSO increase greatly). However, CRO-CPSO can still significantly reduce the journey time of vehicles (25.34%) while maintaining fewer outliers. Therefore, our proposed CRO-CPSO gives full play to the synergy, which not only improves the traffic efficiency of urban road networks, but also fully considers the driving experience of vehicles during road network regulation.

### 4.3. Scalability Analysis

This section is mainly concerned with the scalability of our proposed CRO-CPSO. We focus on the scale of the road network structure and the scalability of CRO-CPSO synergies in a large-scale road network. In this analysis, there were 2577 intersections and 3497 sections in the simulation of a large-scale urban road network. As shown in Figure 10, when the road network scale is greatly expanded, CRO-CPSO can still achieve good convergence and excellent regulation performance. We found that in the last two iterations, the best fitness value was greatly improved. Through analysis, we found that only 3.50% of the vehicles did not arrive at their destinations on time (failing to obtain $D_{\mathrm{score}}$). That is, the regulation strategy optimized by CRO-CPSO was almost able to guarantee the driving experience of all vehicles, which result was close to the optimal regulation strategy. Thus, our proposed CRO-CPSO has very good scalability and can well adapt to super-large-scale road network structures.

## 5. Conclusions

In this study, we designed a traffic signal regulation model based on edge computing for traffic congestion in large urban road networks. We used a traffic flow prediction system based on fuzzy logic to predict short-term real-time vehicle waiting times. The co-factor set was generated by offline learning, and the potential influence of historical traffic flow in adjacent sections was fully considered. We proposed the CRO-CPSO algorithm to effectively adapt to large-scale road network structures and complex traffic conditions. We used SUMO, a well-known microscopic traffic simulator, to evaluate our solution. We tested CRO-CPSO in a modern urban road network scenario of Changsha County and compared it with the classical algorithms GA and PSO. A series of analyses were carried out from different viewpoints, from which the following conclusions can be extracted. Our CRO-CPSO algorithm performs successfully in the generation of optimized scheduling strategies for large-scale realistic traffic scenarios. For all the instances, our proposal obtained robust results which were better than those of the other two algorithms compared: the GA and PSO algorithms. Our suggested algorithm can enhance driving experience and shorten journey times. The performance of CRO-CPSO improved by 2.70% (100 vehicles), 4.10% (300 vehicles) and 13.03% (500 vehicles) with different traffic flow densities. CRO-CPSO can also shorten the journey times of vehicles by 25.34% under conditions of high road congestion. All of this means real improvement in city traffic conditions. Additionally, CRO-CPSO has good scalability. When the scenario was extended to 2577 intersections and 3497 sections, the scheduling strategy could still ensure the driving experience of all drivers and was close to the optimal solution for scheduling. Thus, our method can well adapt to different road network sizes and traffic densities.

In future work, we will further improve the training parameters of the co-factor set to expand the coordination range for road scheduling. We will further optimize the rule setting according to fuzzy logic. We will also consider introducing machine learning methods to predict traffic flow in real time and consider pedestrians and other multi-objective factors in traffic control. A future study will also focus on the emergency treatment of traffic emergency faults so as to further fit real urban road scenarios.

## Figures and Tables

**Figure 1 sensors-22-05953-f001:**
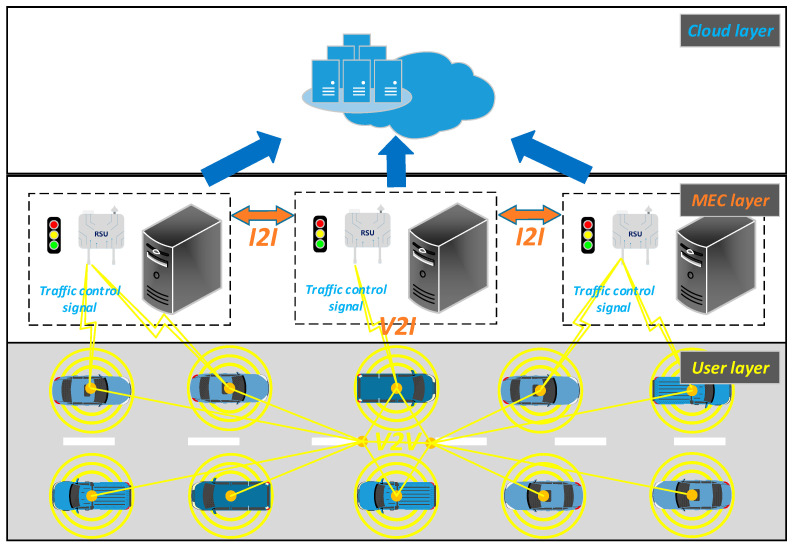
Edge swarm intelligence ATSC environment.

**Figure 2 sensors-22-05953-f002:**
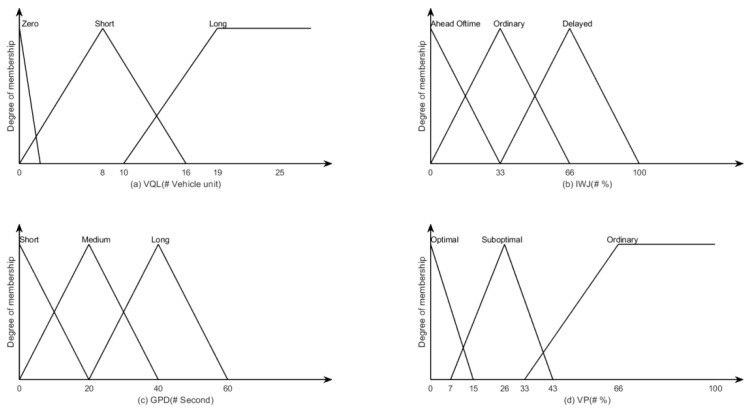
The input fuzzy membership of the fuzzy logic system.

**Figure 3 sensors-22-05953-f003:**
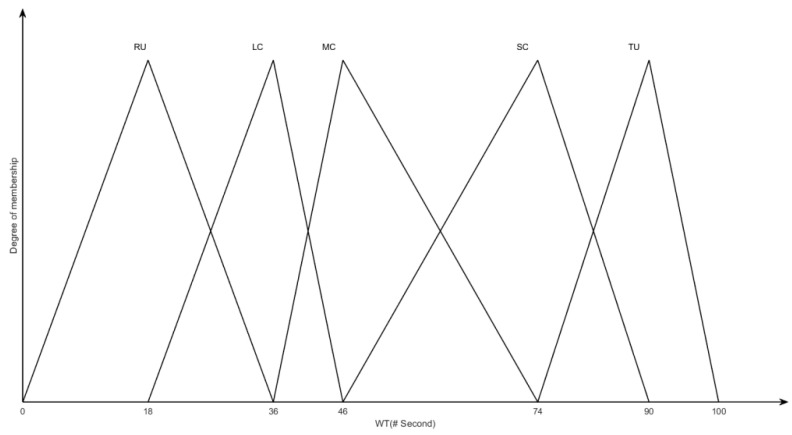
The output fuzzy membership of the fuzzy logic system.

**Figure 4 sensors-22-05953-f004:**
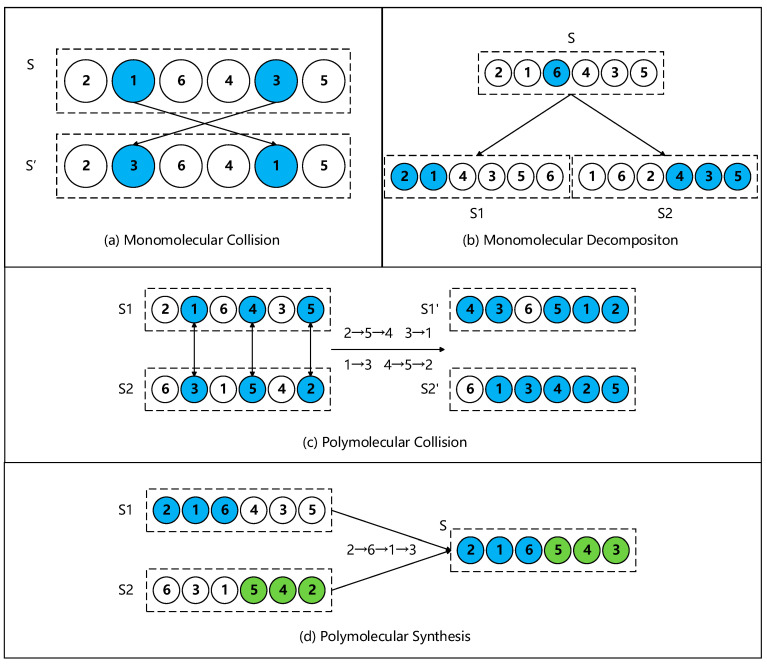
Examples of the updating scheme for the solution space $S{}^{\prime }.$ (The blue and green particles in the figure represent the updated part of the solution space, respectively).

**Figure 5 sensors-22-05953-f005:**
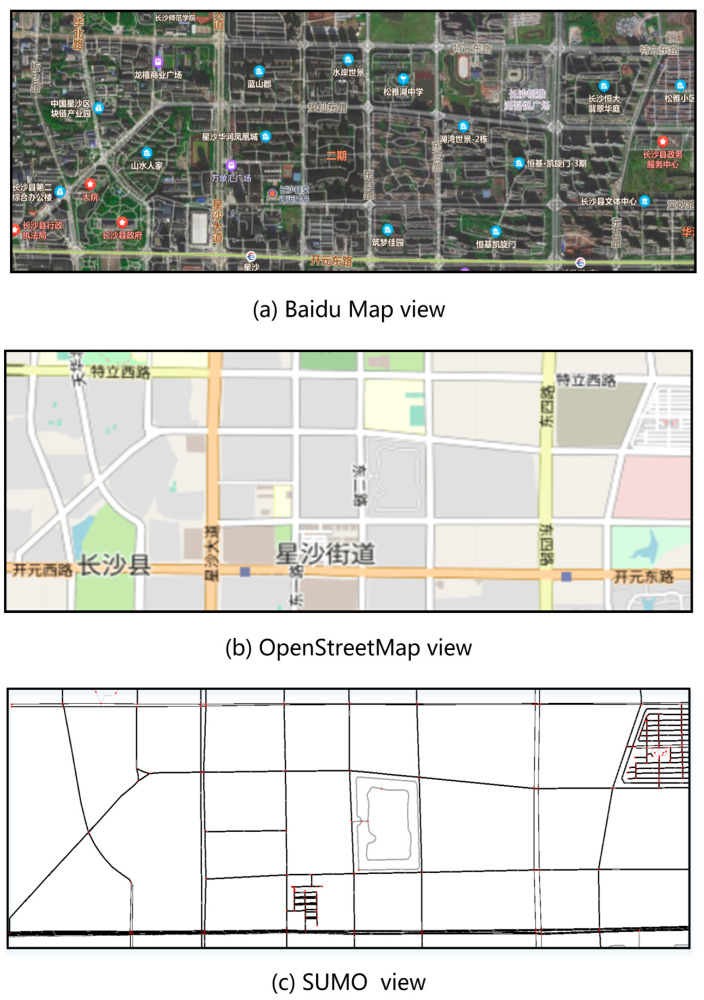
Process of creation of actual road network scenario in the real world. Views of Changsha County (113°6′34.2″ E, 28°15′28.7″ N). After selecting the actual road network scenario (Baidu Map view), we transformed it by the means of the OSM tool and then exported it to SUMO by means of the netconvert script (a script that can convert various third-party road network files into sumo readable files).

**Figure 6 sensors-22-05953-f006:**
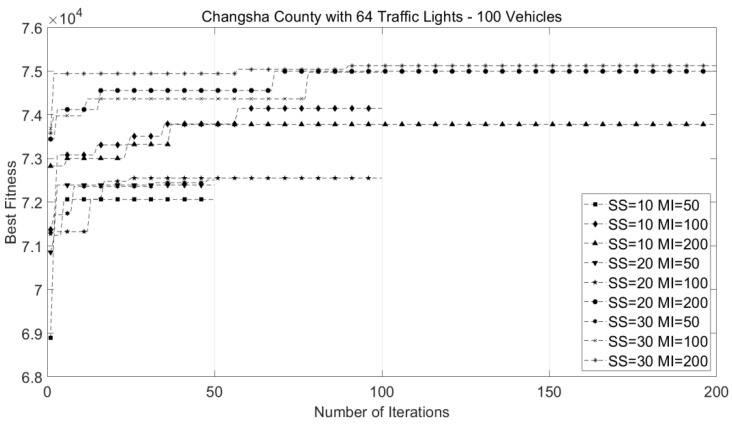
Traces of progress track of best fitness values of CRO-CPSO in the case of Changsha County. SS = swarm size, MI = maximum iteration.

**Figure 7 sensors-22-05953-f007:**
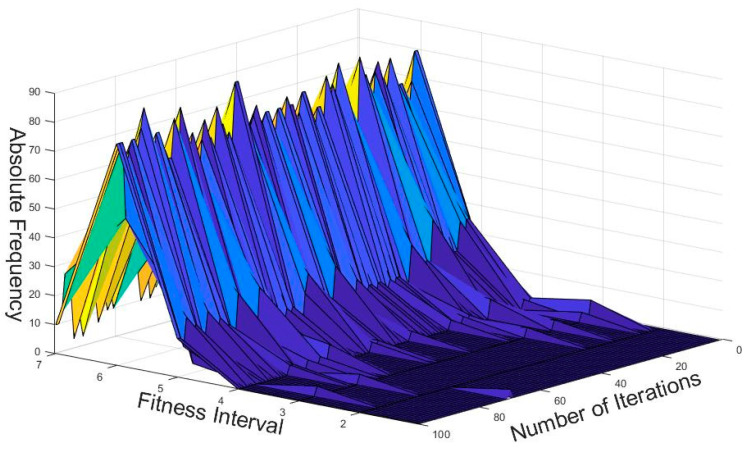
Absolute frequency histogram of swarm fitness.

**Figure 8 sensors-22-05953-f008:**
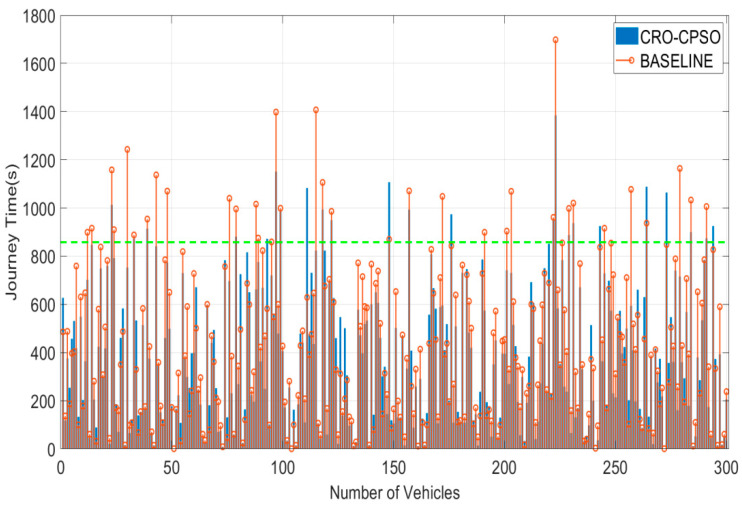
Comparison performance diagram of the driving time of 300 vehicles after CRO-CPSO strategy optimization (the green-dotted line is $T_{\Pi }$).

**Figure 9 sensors-22-05953-f009:**
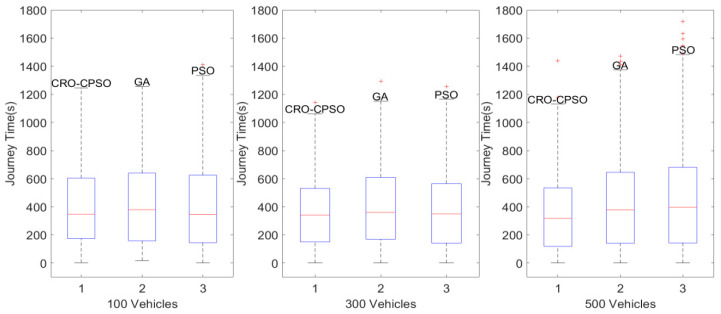
Boxplot representation of distribution results for the journey times of vehicles in Changsha County scenarios with 100, 300 and 500 vehicles.

**Figure 10 sensors-22-05953-f010:**
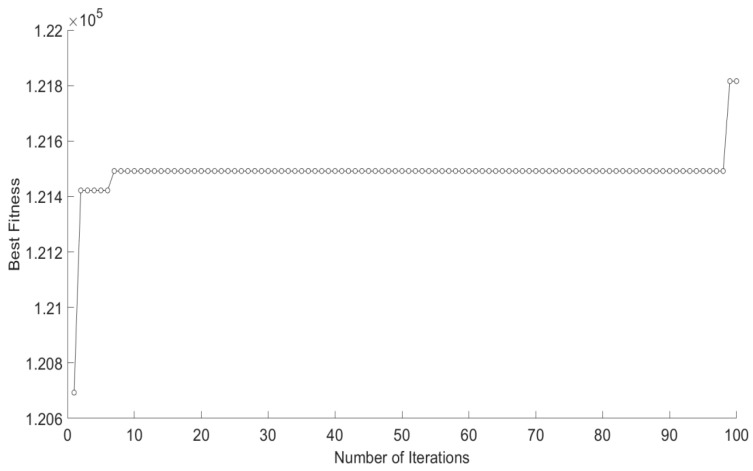
The best fitness trajectory of CRO-CPSO for 2577 intersections and 3497 sections of an urban road network.

**Table 1 sensors-22-05953-t001:** Some rules example of the fuzzy logic.

Rule No	The Rule
Rule 1	If *VQL* is *Short* and *VP* is *Optimal* and *IWJ* is *Ordinary*, then *WT* is *RU*
Rule 2	If *VQL* is *Zero* and *IWJ* is *Ordinary*, then *WT* is *RU*
Rule 3	If *VQL* is *Zero* and *IWJ* is *Delayed*, then *WT* is *SC*
Rule 4	If *VQL* is *Short* and *GPD* is *Short* and *VP* is *Optimal* and *IWJ* is *Delayed*, then *WT* is *LC*
Rule 5	If *VQL* is *Short* and *GPD* is *Short* and *VP* is *Optimal* and *IWJ* is *Delayed*, then *WT* is *SC*
Rule 6	If *VQL* is *Short* and *GPD* is *Short* and *VP* is *Optimal* and *IWJ* is *Delayed*, then *WT* is *SC*
…	…
Rule 42	If *VQL* is *Long* and *GPD* is *Long* and *VP* is *Suboptimal* and *IWJ* is *Delayed*, then *WT* is *SC*
Rule 43	If *VQL* is *Long* and *GPD* is *Long* and *VP* is *Ordinary* an *IWJ* is *Ahead Oftime*, then *WT* is *TU*
Rule 44	If *VQL* is *Long* and *GPD* is *Long* and *VP* is *Ordinary* and *IWJ* is *Ordinary*, then *WT* is *MC*
Rule 45	If *VQL* is *Long* and *GPD* is *Long* and *VP* is *Ordinary* and *IWJ* is *Ahead Oftime*, then *WT* is *TU*

**Table 2 sensors-22-05953-t002:** The detailed simulation parameters.

Simulation Parameters	Value
Simulaiton area	5.14 km^2^
Number of traffic lights	64
Number of vehicles	100/300/500
Vehicle speed	0–20 km/h
$\mathrm{Reward}\;\mathrm{score}\;(D_{\mathrm{score}}$)	250
$\mathrm{Maximum}\;\mathrm{time}\;(T_{\Pi }$)	858 s
$\mathrm{Benchmark}\;\mathrm{distance}\;(d_{\epsilon }$)	1
Number of iterations (P)	30
Swarm size	100
Attenuation factor (α,γ)	0.95
Experiential factor (β)	0.9
$\mathrm{Learning}\;\mathrm{factors}\;(c_{1},c_{2}$)	2
Inertia weight (ω)	0.4

**Table 3 sensors-22-05953-t003:** Best Fitness Values Obtained by CRO-CPSO, GA, and PSO.

Instance	Number of Vehicles	Method		
		CRO-CPSO	GA	PSO
Changsha County	100	**7.05 × 10^3^**	6.95 × 10^3^	6.86 × 10^3^
	300	**2.20 × 10^3^**	2.14 × 10^3^	2.11 × 10^3^
	500	**3.76 × 10^3^**	3.27 × 10^3^	3.35 × 10^3^

The bold is the best fitness value of our CRO-CPSO.

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
