# Peer review of "Coordinated Control of Intelligent Fuzzy Traffic Signal Based on Edge Computing Distribution"

_sensors, 2022, doi:10.3390/s22165953_

Round 1
Reviewer 1 Report
Dear Authors,
First of all, thanks for contributing with this interesting research about control of intelligent traffic signal. After a careful assessment of the paper, I believe the reviewed manuscript addresses a pertinent research problem (i.e. urban road network and traffic saturation) for being considered as publishable in Sensors. Of course, some points need to be addressed beforehand.
In general, should improve the structure of the paper to make it more understandable. It seems that the results are correct and adequate, but the presentation can be clearer and more concise. In this regard, some concerns, queries and suggestions raised during my review must be addressed, in order to optimize the manuscript contents and its suitability for the journal:
· ABSTRACT: Please, explain better or expand information on results. Is important to introduce improvements for traffic saturation (the main problem of your paper) to make it more understandable: our algorithm is more than 13.03% higher than other methods (for what purpose…, climate change, traffic flow, traffic saturation, environmental pollution?
· INTRODUCCTION: After the theoretical justification, you should describe the problem you want to answer and preliminarily raise your research aim and then continue with the paragraph; our study has investigated all of the research gaps mentioned above
· LITERATURE REWIEW: Accurate and well-described
· It would be advisable to reorganize the information in point 4. After counting the experiment and the analysis, enter a results section with the results obtained.
· CONCLUSION: Organize the information in the conclusion and you should introduce a discussion
· DISCCUSSION: please make a discussion and expose (the answers and challenges that your research contributes to road safety and specifically to traffic flow with your model), please study limitations could be improved.
Reviewer 2 Report
The article is interesting and correctly written. The content of the article concerns the development of a traffic control strategy in order to improve traffic efficiency while reducing the rate of road accidents with the use of effective control measures. The aim of the study, resulting from the analysis of the literature, is very well developed. The methods and a well-designed experiment are also properly described.
Remarks:
1. it is worth emphasizing the novelty of the research conducted
2. Please emphasize in the summary whether the developed goals have been achieved?
3. Please complete the list of abbreviations used in the article
4.Figure 4a, 4b ... - worth describing
The revised article may be printed in Sensors.
Reviewer 3 Report
The authors considered the traffic signal control problem in an IoT system based on edge computing. The problem is interesting. The authors proposed a model and algorithm for solving the problem in a large-scale road network. Some detailed comments are as follows:
- The description of the system should be presented more clearly. For example, the data flow in Figure 1 should be described clearly. Where do the proposed algorithms run? What are their input and output? Where do its input parameters come from? Where do its output data go to?
- Problem formulation should be described more clearly. In line 188, the authors stated, "The traffic signal control optimization problem in this paper is a typical multi-objective optimization problem." All objective functions of the problem should be described. All input parameters and output variables of the problem should also be presented clearly.
- The paper presentation should be improved. For example, it would be better if the authors could present the main ideas of CRO-CPSO before going into details. The input and output of Algorithm 2 should be described. The physical meaning of some important parameters should be explained. For example, what is the meaning of T^D_{V_\sigma} (i.e., Eq. (5))? All notations should be summarized in a table.
- The paper should be proofread.
Round 2
Reviewer 3 Report
Thanks for the authors' responses. It would be better if the problem formulation and paper presentation could be improved. For example, as commented in the first version of the paper, in line 296, the authors stated "The traffic signal control optimization problem in this paper is a typical multi-objective optimization problem." However, they introduced only one objective function. It was not easy to follow the paper. The organization and transitions between ideas in a section should be improved. It would be better if the authors could present the main ideas of a section and the proposed algorithms before going into details. The paper should be proofread.
